# Diversity of Mycotoxins and Other Secondary Metabolites Recovered from Blood Oranges Infected by *Colletotrichum*, *Alternaria*, and *Penicillium* Species

**DOI:** 10.3390/toxins15070407

**Published:** 2023-06-21

**Authors:** Ermes Ivan Rovetto, Carlos Luz, Federico La Spada, Giuseppe Meca, Mario Riolo, Santa Olga Cacciola

**Affiliations:** 1Department of Agriculture, Food and Environment, University of Catania, 95123 Catania, Italy; ermes.rovetto@phd.unict.it (E.I.R.); federico.laspada@unict.it (F.L.S.); 2Laboratory of Food Chemistry and Toxicology, Faculty of Pharmacy, University of Valencia, Burjassot, 460100 València, Spain; carlos.luz@uv.es (C.L.); giuseppe.meca@uv.es (G.M.)

**Keywords:** *Alternaria alternata*, *Colletotrichum gloeosporioides*, *Penicillium digitatum*, multilocus phylogenetic analysis, peel, juice, UHPLC–Q-TOF-MS, patulin, rubratoxin B, 5,4-dihydroxy-3,7,8-trimethoxy-6C-methylflavone, atrovenetins

## Abstract

This study identified secondary metabolites produced by *Alternaria alternata*, *Colletotrichum gloeosporioides*, and *Penicillium digitatum* in fruits of two blood orange cultivars before harvest. Analysis was performed by UHPLC–Q-TOF-MS. Three types of fruits were selected, asymptomatic, symptomatic showing necrotic lesions caused by hail, and mummified. Extracts from peel and juice were analyzed separately. *Penicillium digitatum* was the prevalent species recovered from mummified and hail-injured fruits. Among 47 secondary metabolites identified, 16, 18, and 13 were of *A. alternata*, *C. gloeosporioides*, and *P. digitatum*, respectively. Consistently with isolations, indicating the presence of these fungi also in asymptomatic fruits, the metabolic profiles of the peel of hail-injured and asymptomatic fruits did not differ substantially. Major differences were found in the profiles of juice from hail-injured and mummified fruits, such as a significant higher presence of 5,4-dihydroxy-3,7,8-trimethoxy-6C-methylflavone and Atrovenetin, particularly in the juice of mummified fruits of the Tarocco Lempso cultivar. Moreover, the mycotoxins patulin and Rubratoxin B were detected exclusively in mummified fruits. Patulin was detected in both the juice and peel, with a higher relative abundance in the juice, while Rubratoxin B was detected only in the juice. These findings provide basic information for evaluating and preventing the risk of contamination by mycotoxins in the citrus fresh fruit supply chain and juice industry.

## 1. Introduction

The genus *Citrus*, family *Rutaceae*, comprises some of the most widely cultivated fruit crops worldwide, such as oranges, lemons, tangerines, mandarins, limes, grapefruits, and citrons [1,2,3]. Globally, citrus fruits are cultivated in tropical, subtropical, and temperate climates in more than 140 countries and are consumed mainly as fresh fruit or juice [4]. Blood oranges are a group of sweet orange (*Citrus* × *sinensis*) cultivars characterized by a red pigmentation of the rind and flesh, of variable intensity. These oranges are appreciated for their organoleptic characteristics and richness in anthocyanins, a family of polyphenolic pigments. The health benefits of anthocyanins, which possess strong antioxidant properties, include antidiabetic, anticancer, anti-inflammatory, antimicrobial, and anti-obesity effects as well as the prevention of cardiovascular diseases [5]. Like all citrus fruits, blood oranges are a source of other health-promoting substances, such as vitamins, mineral salts, phenolic acids, flavonoids, pectin, and dietary fibers [1,6,7]. The traditional growing areas of blood oranges have long been Sicily, still the leading producer in the world, Spain, Morocco, and Tunisia. As a typical local product, in the EU the red orange from Sicily has Protected Geographical Status. Among the Sicilian blood orange types, the most popular is ‘Tarocco’ which comprises numerous cultivars, differing in intensity of red pigmentation and ripening season, fruit shape and size, tree vigor, and productivity [8]. In recent years, blood oranges have gained vast popularity, and their consumption as fresh fruit or juice has increased and is extending to new countries [9].

Citrus fruits in general are susceptible to infections of fungal pathogens, such as species of *Alternaria*, *Colletotrichum*, *Geotrichum*, and *Penicillium* [10,11,12,13], which are responsible for both pre- and post-harvest rots. The most common and serious post-harvest types of rot of citrus fruits are green and blue molds caused by *Penicillium digitatum* and *P. italicum*, respectively, followed by sour rot caused by *Geotrichum citri-aurantii* [14,15]. These fungi are strict wound pathogens and can infect the fruit in the grove, in the packinghouse, or during subsequent handling and storage. *Colletotrichum gloeosporioides* is responsible for both pre- and post-harvest citrus fruit anthracnose and is the prevalent *Colletotrichum* species associated with citrus worldwide [11,16,17,18]. It resides in citrus fruit peel, supposedly as a saprophyte, endophyte, or epiphyte [19], and can switch to a pathogenic lifestyle under the effect of abiotic stress factors, such as exposure to ethylene during post-harvest fruit degreening or injuries caused by wind, heat, frost, or hail in the field. The precise identification of *Alternaria* species associated with citrus fruits is problematic, as the taxonomy of *Alternaria* is not well defined due to the great morphological and genetic variability of presently recognized species [20,21,22]. *Alternaria alternata*, sensu Woundeberg et al. [20], is prevalent among the small-spored *Alternaria* species associated with brown spot on citrus fruits. Brown spot is a disease affecting mainly tangerines and their hybrids, occurring primarily in citrus production areas with a Mediterranean-type climate [16,23,24,25]. Moreover, it is a prevalent species associated with Alternaria black rot of citrus fruits, also known as core rot [26,27]. Both of these diseases seriously affect the marketability of fresh citrus fruits. Similar to *C. gloeosporioides*, *A. alternata* and other small-spored *Alternaria* species reside in the citrus fruit peel as latent pathogens and can infect and colonize as opportunistic necrotrophic pathogens, injuring fruits of all citrus varieties, including those resistant to the Alternaria brown spot disease.

Fungal pathogens of citrus fruits were reported to produce mycotoxins [3,24,28,29].

The term mycotoxins refers to secondary metabolites toxic for humans and animals produced by filamentous fungi that colonize crop plants as pathogens or saprophytes [28]. Some fungal genera in particular, including *Alternaria*, *Aspergillus*, *Fusarium*, *Claviceps*, and *Penicillium*, are known to produce mycotoxins [29,30,31,32]. The most studied mycotoxins include aflatoxins (AFs), common in dried fruits and nuts; ochratoxin A (OTA), occurring in grapes, raisins, and red wines; and citrinin (CIT), mainly found in stored grains but occasionally also in fruits and other plant products. Mycotoxins, including zearalenone (ZEA), fumonisins (FBs), and *deoxynivalenol* (DON), which is a mycotoxin of the group of trichothecenes (TCTs), are produced by species of *Fusarium* and occur in cereal crops. Moreover, patulin (PAT) is a polyketide mycotoxin produced primarily by *P. expansum* in rotten apples and also occurs in apple juices. Furthermore, ergot alkaloids and indole compounds are produced by various *Claviceps* species on monocotyledonous plants of the families *Poaceae*, *Juncaceae*, and *Cyperaceae*, including forage grasses, corn, wheat, barley, oats, millet, sorghum, rice, and rye. Finally, the dibenzopyrone derivative alternariol (AOH) is produced by the genus *Alternaria* and is a common contaminant of many fruits and vegetables [28,29,33,34,35,36]. Besides AOH, *Alternaria* mycotoxins (ATs) comprise alternariol monomethyl ether (AME) and altenuene (ALT), both of the structural group of the dibenzopyrone derivatives, the perylene derivatives altertoxins (ATX-I, ATX-II, and ATX II), and the tetramic acid derivative tenuazonic acid (TeA) [31,34,37,38,39,40]. Levels of contamination of plant products by mycotoxins may vary depending on several factors, such as cultivar susceptibility, climate, storage condition, damages caused by insect pests, use of pesticides, geographical origin, and mechanical injures due to improper handling or weather events [40,41]. Most mycotoxins are heat stable and resist industrial processes; consequently, they can be found both in fresh fruits and processed products, such as fruit juices [40]. Various mycotoxins have been identified in apple, apricot, berry, orange, peach, and pear juices [31,41,42,43,44,45,46].

Contamination by mycotoxins of plant products and processed food and feed are a serious concern for human and animal health, and several countries have set regulatory limits for mycotoxins in foods and feeds [24,47,48,49]. Moreover, strategies for minimizing contamination by mycotoxins in foods and feeds have been suggested [30]. Finally, actions have been taken to prevent or reduce the risk posed by mycotoxins for human health. In particular, the European Commission (EC) has reinforced border controls for mycotoxins in specific crops imported from given geographical areas [50,51,52]. European and national regulations have established maximum tolerable levels of principal mycotoxins in foods [52]. In fruits and juices, only PAT and OTA are regulated [41]. For PAT, the European Union (EU) [53] set a maximum level of 50 µg/kg in fruit juices, concentrated fruit juices, fruit nectars, cider, and other fermented drinks derived from apples or containing apple juice [3,54]. A limit of 25 µg/kg has been implemented in processed solid apple products (apple compote or apple puree intended for direct consumption) [38,54] and 10 µg/kg for apple-derived products intended for young children and infants [38,54]. For OTA, the EC has set 2 μg/kg as maximum level in reconstituted concentrated grape juice [41], grape nectar [40], grape must, and reconstituted concentrated grape must, intended for direct human consumption [53]. The European Union (EU) has set the maximum acceptable level for Aflatoxin B1 (AFB1) at 2 μg/kg–12 μg/kg (depending on the type of foodstuff) [55]. The EC Recommendation 2006/576/EC5 [56] regulates the presence of DON, ZEA, OTA, FBs, T-2, and HT-2, the last two both of the TCTs group, in products intended for animal feeding [57]. By contrast, even though EFSA (European Food Safety Authority) has published a scientific opinion on the risks deriving for the animal and human health from the presence of ATs in food and feed [38,58], no EU decision has been so far issued to regulate other group of mycotoxins, such as ATs [38,58].

Both the concern and awareness of the risks posed by mycotoxins to health and safety prompted to seek for new robust analytical methods for the extraction and determination of these metabolites in plant products and processed foods [57]. The most common methods for the extraction of mycotoxins in foodstuffs are QuEChERS extraction [41,59], liquid–liquid extraction [41,60], and dispersive liquid–liquid microextraction (DLLME) [40,41,61]. In particular, dispersive liquid–liquid microextraction (DLLME) has showed some advantages, such as simplicity of operation [61], high recoveries, low-cost applications [40,41,61,62], and automation of the analytical procedure [40,63]. This method has been used for the determination of many mycotoxins, for example OTA in wines [61,64], ZEA in beer [64,65], AFs in oils, PAT in apple juices, and AFs and OTA in rice [61]. Tolosa et al. [66] used DLLME for multimycotoxins analysis to evaluate the presence of 15 mycotoxins in water and fish plasma [64]. Rodríguez-Carrasco et al. [67] utilized this method for the analysis of ATs in tomato and tomato-based products, and Serrano et al. [68] used it to investigate enniatin migration from pasta to pasta cooking water.

In recent years, high-performance liquid chromatography quadrupole time-of-flight mass spectrometry (UHPLC–Q-TOF-MS) has been successfully utilized to study changes induced by bio-preservative microorganisms in fungal metabolomic profiles of food-contaminant fungi [69,70,71,72,73,74,75,76,77,78,79,80,81,82,83,84,85], to determine pesticides on tomato peels [86] and to analyze polymethoxylated flavonoids (PMFs) in citrus peels [87] and plants [84].

The aims of this study were (*i*) to apply the UHPLC– Q-TOF-MS analytical method for the characterization of the mycotoxicological profile of mature fruits of two blood orange cultivars, (*ii*) to determine how this profile varies depending on the fungi associated with the fruit peel and in the diverse parts of the fruit, and (*iii*) to investigate how the profile is affected by the phytosanitary status of the fruit as a consequence of an environmental stress, such as hail.

## 2. Results

Approximately 2800 fungal mass isolates were obtained from orange fruits. Mass isolates were preliminarily separated into three groups on the basis of morphotype, i.e., colony morphology on PDA and microscopical traits. The three morphotypes corresponded to *Alternaria*, *Colletotrichum*, and *Penicillium,* respectively. On PDA, *Alternaria* isolates showed flat, woolly colonies ranging from dark green to black in color. They produced typical dark brown, club-shaped conidia arranged in branched chains, with oval-ellipsoidal shapes and 3–5 transverse septa. Colonies of *Colletotrichum* isolates grown on PDA showed a dense, cottony, aerial mycelium that was initially white and turned progressively pale gray with salmon-pink conidial mucilaginous masses in the center of the colony. Dark acervuli were scattered over the entire surface of old colonies, and the colony reverse was pale orange to uniformly grey. The single-celled conidia were hyaline, smooth, cylindrical, with both ends rounded, and dimensions of 11–15 × 4–6 μm. Setae were present in most isolates. On PDA, colonies of *Penicillium* isolates were initially white and then turned green due to abundant production of conidia. The colony margin was entire and narrow, and conidia were borne in chain on typically branched conidiophores; they were globose to subglobose in shape, smooth, with a size range of 2.51–4.22 × 2.35–3.61 μm and mean size (±SD) of 3.38 ± 0.49 × 3 ± 0.36 μm.

Isolates of *Alternaria*, *Colletotrichum*, and *Penicillium* accounted for 12, 28, and 60% of all isolates recovered from orange fruits, respectively, and the proportions of isolates of the three fungal genera in each of the six clusters of fruits are reported in Table 1. 

A set of about 7% of mass isolates of each fungal genus, comprising isolates from all clusters, was randomly selected, and a single-conidium isolate was obtained from each selected mass isolate to be identified molecularly. Overall, 200 isolates, 24 of *Alternaria*, 56 of *Colletotrichum*. and 120 of *Penicillium*, were identified molecularly at species level. In particular, based on the phylogenetic analysis of both the ITS and β-tub regions, the *Colletotrichum* and *Penicillium* isolates were identified as *C. gloeosporioides* and *P. digitatum*, respectively, while based on the phylogenetic analysis of ITS, TEF-1a, and Alt-1a regions, the *Alternaria* isolates were identified as *A. alternata*. As the isolates belonging to the same species were as a result identical, a limited number of sequences of isolates per each species were deposited on Genbank (Table 2).

The analysis by Agilent Ultra High-Definition Accurate Mass revealed 47 diverse fungal secondary metabolites in peel and juice of orange fruits. The identification of the metabolites was supported by using a specifically designed database based on the secondary metabolites produced by species of *Alternaria*, *Colletotrichum*, and *Penicillium* available in the literature (Table 3).

For the calibration of the UHPLC–Q-TOF-MS analysis method, commercial mycotoxins (Sigma-Aldrich) of two of the identified fungi were used as standards, patulin for *P. digitatum* and AME for *A. alternata*, respectively. Precision was performed by inter- and intraday variations to make the analytical method accurate. Mean recoveries were operated on orange juice and orange peel (non-contaminated) using three repetitive injections of selected standards in a day and three consecutive days for the interday variation at levels of patulin and AME (0.01, 0.1 and 1 mg/L for both toxins) which were of 81.1 ± 1.9% and 89.4 ± 2.4% for orange juice and 84.9 ± 2.6% and 92.6 ± 3.1% for orange peel. The results were expressed as relative standard deviations (RSD) of intra-day and interday variation with values ranging between 3.7 and 4.1% for patulin and between 2.9 and 4.5% for AME, respectively. The matrix effects (% MEs) evidenced for patulin and AME in orange juice were 89% and 92%, whereas in orange peel the percentages of the MEs detected were 96% and 98%, respectively.

The secondary metabolites detected by UHPLC-Q-TOF-MS in each fruit cluster are presented as heat maps (Figure 1, Figure 2 and Figure 3). Only secondary metabolites with a large gap between the asymptomatic fruit cluster of each orange cultivar and the clusters of both hail-injured and mummified fruits of the same cultivar are reported in the maps.

Figure 1A shows that the secondary metabolites of *A. alternata* present in orange peel included SM2, SM3, SM4, SM5, SM6, SM9, SM10, SM11, SM12, and SM15. However, only SM3 (AF-toxins II) and SM6 (Alternethanoxin A) were found with a high relative abundance compared to the other metabolites. Specifically, SM3 was found with a high relative abundance in all clusters except for the mummified fruits of both varieties, while SM6 was relatively abundant in all clusters, except in Mum_L.

SM2 (AAL-toxin TE2) was present only in the (Mum_T) cluster; in the (Mum_L) cluster, SM5 (Alternariol monomethyl ether, AME) had a relative abundance, approximately six-fold higher than the (Asym_L) cluster. In addition, in the (H-Inj_T) and (Mum_T) clusters, this metabolite had relative abundance approximately three-fold and nine-fold higher than the (Asym_T) cluster. Alternariol monomethyl ether (AME) (SM5) was reported as cytotoxic activity [90]. In the (Mum_T) cluster, SM6 had relative abundance approximately two-fold higher than in the (Asym_T) cluster. Alternethanoxin A (SM6) displayed in vitro growth inhibitory activity with respect to several cancer cell lines [123]. Furthermore, SM10 (Dihydroaltersolanol) and SM11 (Erythroglaucin) in (H-Inj_L) had relative abundance approximately two-fold and three-fold higher than in (Asym_L).

Figure 1B shows that secondary metabolites released by *A. alternata* in orange juice were SM1, SM2, SM4, SM5, SM6, SM7, SM8, SM9, SM10, SM13, SM14, and SM16. 

However, only SM2, SM6, and SM14 (Porriolide) were found to be relatively abundant compared to the other metabolites. Specifically, in mummified fruits of both orange cultivars, the (Mum_L) and (Mum_T) clusters, SM2 showed a relative abundance around four- and three-fold higher, respectively, than in asymptomatic fruits, the (Asym_L) and (Asym_T) clusters. In hail-injured fruits of both orange cultivars, the (H-Inj_L) and (H-Inj_T) clusters, SM4 (Altenusin) showed a relative abundance around three- and six-fold higher than in asymptomatic fruits, the (Asym_L) and (Asym_T) clusters, respectively. In mummified fruits of both orange cultivars, the (Mum_L) and (Mum_T) clusters, this metabolite showed a relative abundance about 251- and 63-fold higher than in asymptomatic fruits, the (Asym_L) and (Asym_T) clusters, respectively. In the (Mum_L) and (Mum_T) clusters, SM5 showed a relative abundance about four- and two-fold higher than in the (Asym_L) and (Asym_T) clusters, respectively. In the (Mum_L) and (Mum_T) clusters, SM6 (Alternethanoxin A) showed a relative abundance about two-fold higher than in the (Asym_L) and (Asym_T) clusters, respectively. SM9 (Curvularin) was present only in the (Mum_L) and (Mum_T) clusters, while SM10 (Dihydroaltersolanol) in the (H-Inj_L) and (Mum_L) clusters showed a relative abundance approximately two- and five-fold higher, respectively, than in (Asym_L). SM14 (Porriolide) was recorded with a high relative abundance in (Asym_T) and (H-Inj_T) compared with other clusters.

Figure 2A,B show the relative abundance of secondary metabolites of *Colletotrichum* produced in the peel and juice of orange fruits. The following metabolites were detected in the peel: SM19, SM20, SM21, SM22, SM24, SM25, SM27, SM28, SM29, SM30, SM31, SM32, and SM33. Among them, SM19 (5,4-dihydroxy-3,6,7-trimethoxy-8C-methylflavone) showed a higher relative abundance in the hail-injured fruits, the (H-Inj_L) cluster, compared to the asymptomatic fruits, the (Asym_L) cluster. Similarly, the (H-Inj_T) and (Mum_T) clusters showed a higher relative abundance in SM19 (5,4-dihydroxy-3,6,7-trimethoxy-8C-methylflavone) than in the (Asym_T) cluster. SM20 (5,4-dihydroxy-3,7,8-trimethoxy-6C-methylflavone) and SM22 (Apigenin-8-C-β-D-glucopyranoside) were found with a high relative abundance in all clusters of both varieties. SM21 (Alternariol-5-Me ether) showed a notable higher relative abundance in the (Mum_L) cluster compared to the (Asym_L) cluster. SM21 (Alternariol-5-Me ether) showed a higher relative abundance in the (H-Inj_T) and (Mum_T) clusters compared to the (Asym_T) cluster. SM25 (Colletoic acid) was detected at higher levels in both the (Mum_L) and (Mum_T) clusters compared to the (Asym_L) and (Asym_T) clusters. In the (Mum_L) and (Mum_T) clusters, SM29 (Colletotrilactam A) was detected with a higher relative abundance than in other clusters.

Figure 2B shows the relative abundance of secondary metabolites produced by *C. gloeosporioides* in orange juice, including SM17, SM18, SM19, SM20, SM21, SM22, SM23, SM24, SM25, SM26, SM29, SM30, SM32, and SM34. Among them, SM18 (Ergosterol peroxide) was only present in mummified fruits, the (Mum_L) and (Mum_T) clusters. SM19 (5,4-dihydroxy-3,6,7-trimethoxy-8C-methylflavone) was present at higher levels in the (H-Inj_L) and (Mum_L) clusters compared to the (Asym_L) cluster and in the (H-Inj_T) and (Mum_T) clusters compared to the (Asym_T) cluster. SM20 (5,4-dihydroxy-3,7,8-trimethoxy-6C-methylflavone), SM22 (Apigenin-8-C-β-D-glucopyranoside), and SM25 (Colletoic acid) showed a much higher relative abundance in the (Mum_L) and (Mum_T) clusters compared to the (Asym_L) and (Asym_T) clusters. However, SM24 (Colletofragarone A1) was found with a high relative abundance in all clusters except for the mummified fruits of both orange cultivars. SM25 (Colletoic acid) showed a much higher relative abundance in the (Mum_L) and (Mum_T) clusters compared to the (Asym_L) and (Asym_T) clusters. Finally, SM23 (Collectotrichin A) and SM34 (Pyrenocine A) were only present in the (Mum_L) and (Mum_T) clusters and were not detected in the (Asym_L) and (Asym_T) clusters.

Figure 3A,B show the relative abundance of *Penicillium* metabolites detected in peel and juice of orange fruits. The secondary metabolites produced by *P. digitatum* in the in orange peel were SM35, SM36, SM37, SM38, SM39, SM40, SM42, SM43, SM44, SM45, SM46, and SM47.

However, in the orange peel, only SM35 (Alantrypinone) and SM38 (Atrovenetins) were found with a high relative abundance compared to the other metabolites. Specifically, these secondary metabolites were found with a high relative abundance in all clusters of both orange cultivars. In cluster (H-Inj_L), SM36 (Anacine) was nine-fold more abundant than in cluster (Asym_L). SM37 (Asteltoxin), SM39 (Fungisporin), SM44 (Rubratoxin B), and SM47 (patulin) were present only in mummified fruits of both orange cultivars, i.e., in the (Mum_L) and (Mum_T) clusters. SM42 (Penipacid B) was found with a high relative abundance in all clusters except for the mummified fruits of both orange cultivars. SM43 (Penochalasin K) was detected in hail-injured and mummified fruits, the (H-Inj_L) and (Mum_L) clusters, but was almost absent in asymptomatic fruits, cluster (Asym_L). In hail-injured and mummified fruits of ‘Tarocco Tapi’, the (H-Inj_T) and (Mum_T) clusters, respectively, this metabolite was about 2- and 63-fold, respectively, more abundant than in asymptomatic fruits, cluster (Asym_T). Finally, SM46 (Solistatin), in hail-injured fruits of both ‘Tarocco Lempso’ and ‘Tarocco Tapi’, clusters (H-Inj_L) and (H-Inj_T), was about 8- and 15-fold, respectively, more abundant than in asymptomatic fruits of both cultivars, clusters (Asym_L) and (Asym_T), respectively.

The secondary metabolites detected in orange juice using the *Penicillium* dataset were SM35, SM38, SM39, SM40, SM41, SM43, SM44, SM45, and SM47. Specifically, SM35 (Alantrypinone) and SM38 (Atrovenetins) were found with a higher relative abundance compared to the other metabolites. In juice of mummified fruits of both orange cultivars, the (Mum_L) and (Mum_T) clusters, the relative abundance of SM35 (Alantrypinone) was approximately four- and three-fold higher than in juice of asymptomatic fruits of both cultivars, the (Asym_L) and (Asym_T) clusters, respectively. Similarly, the relative abundance of SM38 (Atrovenetins) in mummified fruits of both orange cultivars, the (Mum_L) and (Mum_T) clusters, was approximately four- and three-fold higher than in asymptomatic fruits, the (Asym_L) and (Asym_T) clusters, respectively. In the (Mum_L) cluster, SM39 (Fungisporin) showed a relative abundance approximately 355-fold higher than in cluster (Asym_L); it was present in the (Mum_T) cluster but was not detected in cluster (Asym_T). SM40 (Lichexanthone) was found in cluster (Mum_L) but not in cluster (Asym_T). Moreover, this metabolite in (Mum_T) showed a relative abundance approximately 13-fold higher than in cluster (Asym_T). SM43 (Penochalasin K), in mummified fruits of ‘Tarocco Lempso’, cluster (Mum_L), showed a relative abundance approximately 11-fold higher than in asymptomatic fruits of the same cultivar, cluster (Asym_L). This metabolite was also detected in mummified fruits of ‘Tarocco Tapi’, cluster (Mum_T), but was not found in asymptomatic fruits of this orange cultivar, cluster (Asym_T). SM44 (Rubratoxin B) and SM47 (patulin) were found only in the (Mum_L) and (Mum_T) clusters. In mummified fruits of the two orange cultivars, clusters (Mum_L) and (Mum_T), SM45 (Serantrypinone) was detected with a relative abundance about five- and four-fold higher than in asymptomatic fruits of the respective cultivars, clusters (Asym_L) and (Asym_T).

Principal Component Analysis (PCA) of data was performed. It was based on the secondary metabolites of *A. alternata*, *C. gloeosporioides*, and *P. digitatum* (Table 2) identified in the peel and juice of asymptomatic, hail-injured, and mummified fruits of the two blood orange cultivars, ‘Tarocco Lempso’ and ‘Tarocco Tapi’.

Figure 4 shows the clustering of asymptomatic, hail-injured, and mummified orange fruits of the two orange cultivars in the score plot and loading plot, based on secondary metabolites of *Alternaria* detected in the peel. The sum of the principal component values accounted for 69.9% of the total variance, with PC1 representing 47.3% and PC2 representing 22.6% of the total variance.

In the score plot, asymptomatic orange fruits (Asym_T and Asym_L) were clustered in quadrants II and IV, while hail-injured fruits (H-Inj_T and H-Inj_L) were clustered in quadrant II. Mummified fruits of ‘Tarocco Lempso’ (Mum_L), with the only exception of replicate #1, tended towards quadrant IV. Conversely, mummified fruits of ‘Tarocco Tapi’ (Mum_T) were clustered in quadrant III.

In the loading plot, the hail-injured fruits (H-Inj_T and H-Inj_L) were clustered in quadrant I, characterized by the secondary metabolites SM3, SM4, SM6, SM9, SM11, SM12, and SM15. By contrast, the mummified fruits of ‘Tarocco Tapi’ were clustered in quadrant I, characterized by secondary metabolites SM2 and SM5 (Figure 4b).

The score plot and loading plot of Figure 5 illustrate the clustering of the three types of fruits (asymptomatic, hail-injured, and mummified) of ‘Tarocco Lempso’ and ‘Tarocco Tapi’ based on secondary metabolites of *Colletotrichum* detected in the peel. The principal component values accounted for 72.2% of the total variance, with PC1 and PC2 accounting for 58.6% and 13.6%, respectively.

In the score plot, the asymptomatic fruits of the two orange cultivars (Asym_L and Asym_T) were grouped in quadrant II and IV, respectively. The hail-injured fruits of ‘Tarocco Lempso’ (H-Inj_L) were clustered in quadrant III, while the hail-injured fruits of ‘Tarocco Tapi’ (H-Inj_T) were spread across quadrants I, II, and IV. The mummified fruits of ‘Tarocco Lempso’ and ‘Tarocco Tapi’ (Mum_L and Mum_T) were clustered in quadrants III and I, respectively, with the only exception of Mum_T replicate #1, which tended towards quadrant IV.

The loading plot showed that secondary metabolites associated with the peel of mummified fruits of ‘Tarocco Lempso’ (Mum_L) clustering in quadrant I included SM1, SM2, SM3, SM4, SM6, SM9, and SM12, while secondary metabolites associated with the peel of hail-injured fruits of ‘Tarocco Tapi’ (H-Inj_T), clustering in quadrant I, included SM5, SM7, SM8, SM10, SM11, and SM15.

In the loading plot, secondary metabolites SM19, SM22, SM24, SM27, SM28, SM30, SM31, SM32, and SM33 tended towards quadrants I and III, clustering the hail-injured fruits of ‘Tarocco Lempso’ (H-Inj_L) as well as the mummified fruits of both orange cultivars (Mum_L and Mum_T) (Figure 5).

Figure 6 shows the clustering of asymptomatic, hail-injured, and mummified fruits of ‘Tarocco Lempso’ and ‘Tarocco Tapi’ on the basis of *Penicillium* secondary metabolites detected in the peel (score plot) and the metabolite trends (loading plot) in different quadrants (I, II, III, and IV); the sum of the principal component values accounted for 69.7% of the total variance. PC1 represented 51.1% and PC2 18.6% of the total variance.

The score plot of *Penicillium* metabolites detected in the peel shows that asymptomatic fruits (Asym_L and Asym_T) were clustered in quadrant IV, with the exception of replicate #3, which tended towards quadrant II. Hail-Injured fruits of ‘Tarocco Lempso’ (H-Inj_L) were clustered in quadrant I, while fruits of ‘Tarocco Tapi’ (H-Inj_T) were scattered in quadrants I, II, and IV. The mummified fruits of both cultivars (Mum_T and Mum_L) were clustered in quadrant III, with the exception of cluster (Mum_T) replicate # 1, which tended towards quadrant IV.

In the loading plot, secondary metabolites SM35, SM36, SM37, SM39, SM40, SM42, SM43, SM44, SM45, SM46, and SM47 tended towards quadrants I and III, clustering the samples of peel of hail-injured fruits of ‘Tarocco Lempso’ (H-Inj_L), replicate #3 of hail-injured fruits of ‘Tarocco Tapi’ (H_Inj_T_3), and mummified fruits of both orange cultivars (Mum_L and Mum_T) (Figure 6).

Figure 7 shows the clustering of asymptomatic, hail-injured, and mummified orange fruits of ‘Tarocco Lempso’ and ‘Tarocco Tapi’ based on *Alternaria* metabolites (score plot) detected in the juice and the metabolite trends (loading plot) in different quadrants (I, II, III, and IV). The sum of the principal 12 component values reached 76.3% of the total variance. PC1 and PC2 accounted for 48.9 and 27.4% of the total variance, respectively.

In the score plot of *Alternaria alternata*, asymptomatic fruits of the two orange cultivars (Asym_T and Asym_L) were clustered in quadrants I and III, respectively, with the only exception of replicate #3 of asymptomatic fruits of ‘Tarocco Tapi’ (Asym_T_3), which tended towards quadrant III. Hail-injured fruits of ‘Tarocco Lempso’ (H-Inj_L) were clustered in quadrant III, while hail-injured fruits of ‘Tarocco Tapi’ (H-Inj_T) were clustered in quadrant I, with the only exception of replicate #3 (H-Inj_T_3) that tended towards quadrant III. Mummified fruits of ‘Tarocco Lempso’ (Mum_L) were clustered in quadrant II, while mummified fruits of ‘Tarocco Tapi’ (Mum_T) were clustered in quadrants I, II, and IV. 

In the loading plot, the secondary metabolites SM1, SM7, SM8, SM13, and SM16 tended towards quadrant II, clustering the mummified fruits of ‘Tarocco Lempso’ (Mum_L and Mum_T), and the metabolites SM2 SM4, SM5, SM6, SM10, and SM14 clustered the hail-injured fruits of both orange cultivars (H-Inj_L and Mum_T) (Figure 7).

Figure 8 shows the clustering of asymptomatic, hail-injured, and mummified fruits of ‘Tarocco Lempso’ and ‘Tarocco Tapi’, based on *Colletotrichum* secondary metabolites detected in the juice (score plot) and the metabolite trends (loading plot) in different quadrants (I, II, III, and IV). The sum of the principal 14 component values accounted for 70.3% of the total variance. PC1 and PC2 accounted for 50.9 and 19.4% of the total variance, respectively.

In the score plot of *Colletotrichum gloeosporioides*, asymptomatic fruits of both orange cultivars (Asym_T and Asym_L) were clustered in quadrants II and IV, respectively, with the only exception of replicate #3 of ‘Tarocco Tapi’ (Asym_T_3), which tended towards quadrant IV. Hail-injured fruits of both orange cultivars (H-Inj_L and H-Inj_T) were clustered in quadrant III, with the only exception of replicate #1 of hail-injured ‘Tarocco Tapi’ (H-Inj_T_1), which tended towards quadrant II. Mummified fruits of both orange cultivars (Mum_L and Mum_T) were clustered in quadrant I, with the only exception of replicate #1 of mummified ‘Tarocco Tapi’ (Mum_T_1), which tended towards quadrant II. In the loading plot, the secondary metabolites, SM18, SM19, SM23, SM29, SM30, and SM34, tended towards quadrant I, clustering the mummified fruits of both orange cultivars (Mum_L and Mum_T), while the secondary metabolites SM17, SM20, SM24, SM25, SM26, and SM32 tended towards quadrant III, clustering the hail-injured fruits of both orange cultivars (H-Inj_L and H-Inj_T) (Figure 8).

Figure 9 shows the clustering of asymptomatic, hail-injured, and mummified fruits of ‘Tarocco Lempso’ and ‘Tarocco Tapi’, based on *Penicillium* secondary metabolites detected in the juice (score plot) and the metabolite trends (loading plot) in different quadrants (I, II, III, and IV). The sum of the principal nine component values accounted for 77.2% of the total variance. PC1 and PC2 accounted for 60.2 and 17% of the total variance, respectively.

In the score plot of *Penicillium digitatum*, asymptomatic fruits of the two orange cultivars (Asym_T and Asym_L) were clustered in quadrants I and III. Hail-injured fruits of both orange cultivars (H-Inj_L and H-Inj_T) were clustered in quadrants I and II, while mummified fruits of the two cultivars (Mum_L and Mum_T) were clustered in quadrants III and IV. In the loading plot, the secondary metabolites SM38, SM39, SM41, and SM45 tended towards quadrant II, clustering the hail-injured fruits of both ‘Tarocco Lempso’ and ‘Tarocco Tapi’ (H-Inj_L and H-Inj_T), while the secondary metabolites SM40 and SM47 tended towards quadrant IV, clustering the mummified fruits of both orange cultivars (Mum_L and Mum_T) (Figure 9).

To summarize, the metabolic profiles of peel extracts of ‘Tarocco Lempso’ and ‘Tarocco Tapi’ and the metabolic profiles of the three types of fruits overlapped only in part, as shown by Ven diagrams (Figure 10). In particular, the secondary metabolites Colletotrilactam (SM29) and patulin (SM47) were produced exclusively in the peel of mummified fruits of both cultivars. In contrast, all clusters shared the secondary metabolites Alternethanoxin A (SM6), 5,4-dihydroxy-3,7,8-trimethoxy-6C-methylflavone (SM20), apigenin-8-C-β-D-glucopyranoside (SM22), alantrypinone (SM35), and Atrovenetins (SM38). AF-toxins II (SM3) and Penipacid B (SM42) were found in the peel of hail-injured fruits of both cultivars, while Colletomelleins B (SM27) and Fusarentin 6,7-dimethyl ether (SM 31) were found exclusively in the peel of hail-injured fruits of ‘Tarocco Lempso’ and ‘Tarocco Tapi’, respectively. Finally, Serantrypinone (SM45) was detected in the peel of both hail-injured and asymptomatic fruits but only in fruits of ‘Tarocco Lempso’.

The metabolic profiles of the juice from mummified fruits of both ‘Tarocco Lempso’ and ‘Tarocco Tapi’ cultivars (Figure 11) had in common several secondary metabolites, including AAL-toxin TE2 (SM2), Alternethanoxin A (SM6), ergosterol peroxide (SM18), apigenin-8-C-β-D-glucopyranoside (SM22), Colletoic acid (SM25), Rubratoxin B (SM44), and patulin (SM47). However, while SM2, SM18, SM22, SM25, SM44, and SM47 were found exclusively in the profile of mummified fruits, SM6 was found in the profile of all types of fruits. Similarly, Alantrypinone (SM35) was found in all types of fruits but only in those of ‘Tarocco Lempso’, while 5,4-dihydroxy-3,7,8-trimethoxy-6C-methylflavone (SM20) and Atrovenetins (SM38) were found in all types of fruits of ‘Tarocco Tapi’ and only in hail-injured and mummified fruits of ‘Tarocco Lempso’. The metabolic profiles of asymptomatic and hail-injured fruits of both orange cultivars shared Porriolide (SM14) and Colletofragarone A1 (SM24), while (+)-(3R,4S)-cis-4-hydroxy-6-deoxyscytalone was detected exclusively in the profile of asymptomatic fruits of ‘Tarocco Tapi’.

## 3. Discussion

In this study, isolations with a conventional microbiological method from the fruit peel of two blood orange cultivars, ‘Tarocco Lempso’ and ‘Tarocco Tapi’, sampled in commercial citrus orchards of a typical production area of eastern Sicily, yielded three species of fungi, *A. alternata*, *C. gloeosporioides*, and *P. digitatum*. Fruits of each cultivar were separated into three distinct groups on the basis of external symptoms, i.e., fruits with the rind injured by the hail, mummified fruits, and asymptomatic fruits, with no visible lesions of the rind. Then, secondary metabolites of peels and juices from the three groups of fruits were extracted and analyzed by UHPLC–Q-TOF-MS. *Penicillium digitatum* was the prevalent species recovered from mummified and hail-injured fruits. The UHPLC-Q-TOF-MS analysis revealed 47 secondary metabolites, of which 16, 18, and 13 were specifically associated with *A. alternata*, *C. gloeosporioides*, and *P. digitatum*, respectively. As regards the metabolites recovered from the three groups of fruits, major differences were found in the profiles of juice from hail-injured and mummified fruits, with a significant higher presence of 5,4-dihydroxy-3,7,8-trimethoxy-6C-methylflavone and Atrovenetin in the juice of mummified fruits of the Tarocco Lempso cultivar. Moreover, the mycotoxins patulin and Rubratoxin B were detected exclusively in mummified fruits. Patulin was detected in both the juice and peel, with a higher relative abundance in the juice, while Rubratoxin B was detected only in the juice.

The prevalent isolation of *Penicillium digitatum* from both mummified and hail-injured fruits of both orange cultivars suggests this fungus had a prominent role in fruit mummification and confirms its nature of being a wound parasite. Both *C. gloeosporioides* and *A. alternata* were recovered from necrotic lesions of hail-injured fruits of both orange cultivars in a significant higher frequency than from the intact rind of asymptomatic fruits. These fungi are ubiquitous in citrus orchards and normally reside in the peel of citrus fruits as epiphytes, endophytes, or latent pathogens. Very probably, in fruits injured by hail, their populations increased because they colonized the rind lesions as opportunistic necrotrophic pathogens. A possible explanation of both the failure in isolating *A. alternata* and the low proportion of successful isolations of *C. gloeosporioides* from mummified fruits is that *P. digitatum* exerted an antagonistic activity against the other two fungal species, based on the competition for the substrate or the production of fungitoxic secondary metabolites by this fungus. *Alternaria alternata*, *C. gloeosporioides*, and *P. digitatum* are responsible for some of the most common and severe pre- and post-harvest diseases of citrus fruits worldwide [1,13,23]. Moreover, they belong to three among the most known genera of mycotoxigenic fungi [58,124]. Overall, *Alternaria* species produce more than 70 mycotoxins sensu lato [94], and more than 100 secondary metabolites of *Colletotrichum* species have been reported so far [103]. *Penicillium* species produce a wide range of mycotoxins of concern for human and animal health. Among mycotoxins occurring in foods and feeds contaminated by *Penicillium* species, the most important are ochratoxin A and patulin, which are regulated in a number of countries, and to a lesser extent cyclopiazonic acid [124]. The identification of culturable fungi recovered from the peel of ‘Tarocco Lempso’ and ‘Tarocco Tapi’ fruits oriented the design and development of *Alternaria*, *Colletotrichum*, and *Penicillium* metabolite databases, that in this study were essential tools for the exploitation of UHPLC–Q-TOF-MS analysis method. Very recently, this innovative analytical approach has been successfully used for identifying the secondary metabolites produced by diverse *Colletotrichum* species both in vitro and on olives infected by anthracnose [125]. Actually, in the present study the combined application of UHPLC–Q-TOF-MS and databases of fungal metabolites proved to be very effective in identifying the secondary metabolites released by *Alternaria*, *Colletotrichum*, and *Penicillium* in the peel and juice of fruits of ‘Tarocco Lempso’ and ‘Tarocco Tapi’. Overall, as many as 47 diverse fungal secondary metabolites were identified in the peel and juice of fruits of these two blood orange cultivars. Of these metabolites, 16 were produced by *A. alternata*; 18 were produced by *C. gloeosporioides*, and 13 were produced by *P. digitatum*. According to the scientific literature, all these metabolites are functionally active, and most of them show diverse types of activity. However, not all bioactive secondary metabolites produced by *A. alternata*, *C. gloeosporioides*, and *P. digitatum* detected in the peel and juice of citrus fruits in this study can be regarded as mycotoxins, sensu stricto [28,126].

In particular, as for the metabolites of *Alternaria*, Alternariol monomethyl ether (AME) is a mycotoxin that can cause kidney damage and immunosuppression in humans and animals [40,102,127]. AME was also reported to induce mitochondrial apoptosis in human colon carcinoma cells [90]. Alternethanoxin A displayed an in vitro inhibitory activity of the growth of cancer cells [123]. Aurasperone-C is known to cause nervous system dysfunctions [128]. Curvularin, which in this study was found only in mummified fruits, was reported to be a carcinogenic and immunotoxic substance [129]. Altenusin (ALN) was reported to be a multifunctional metabolite, with antibacterial, antifungal, and antiparasitic activity [130]. The amount of this metabolite in orange fruits varied considerably. In juice of hail-injured fruits, its relative abundance was 3- to 6-fold higher than in asymptomatic fruits while in the juice of mummified fruits its relative abundance was even 63- to 251-fold higher than in asymptomatic fruits. AAL-toxins were demonstrated to have phytotoxic, cytotoxic, and genotoxic activity [88,131]. AAL-toxin TE2 was found with a high relative abundance in the juice of mummified orange fruits of both cultivars. This metabolite belongs to the AAL-toxins, a group of mycotoxins produced by *A. alternata* structurally related to Fuminosins produced by *Fusarium* species. AAL-toxins were reported as contaminants of various crops and as host specific phytoxins (HSTs) [94,132]. In the *A. alternata*/tomato pathosystem, they have a key role as pathogenicity factors [88,133,134]. AF-toxin II is a mixture of three stereoisomers reported to be phytotoxic on strawberry and pear fruits [134]. In this study, AF-toxin II is reported for the first time on citrus fruits infected by *A. alternata*. This metabolite was detected only in the peel of asymptomatic and hail-injured fruits of both orange cultivars examined. Maculosin was proposed as a potential eco-friendly agrochemical with herbicidal activity [94], while Porriolide was shown to exert a significant antifungal activity [135]. In this study, Alternethanoxin A was found with a high relative abundance in all types of orange fruits, including asymptomatic, hail-injured, and mummified fruits. In the original description of this metabolite isolated from *A. sonchi* cultures, it was envisaged the possibility of using it as a mycoerbicide in view of its broad-spectrum phytotoxic activity [136].

As for *Colletotrichum* metabolites identified in this study, Colletofragarone A1, Colletoic acid, and Pyrenocine A were reported to have phytotoxic and, in a few cases, cytotoxic properties [137,138]. Other compounds such as 5,4-dihydroxy-3,7,8-trimethoxy-6C-methylflavone and Apigenin-8-C-β-D-glucopyranoside, which in this study were found with a high relative abundance in all types of orange fruits examined, were reported in the literature as bioactive metabolites [101,103]. Apigenin, in particular, was demonstrated to have a broad-spectrum anticancer activity [103,139]. However, none of these *Colletotrichum* metabolites have been so far reported as a mycotoxin of concern for human and animal health.

As for *Penicillium*, it is known this fungus produces a wide range of secondary bioactive metabolites, including metabolites with cytotoxic activity such as Penicillic acid, patulin, and Ochratoxin A [124,140]. Patulin is a mycotoxin that has been found as a contaminant in various types of fruits, including oranges [54]. It has been identified as a potential hazard for human health and food safety [18]. Several studies evidenced patulin can cause adverse health effects in humans, including cytotoxicity, genotoxicity, and immunotoxicity [42,141]. The European Union and the United States Food and Drug Administration have set maximum limits for patulin in food products, including oranges [26].

In this study, consistently with the results of isolations, high levels of patulin were detected in mummified fruits of both orange cultivars, in particular in juice. Rubratoxin B was another secondary metabolite found with a high relative abundance in the juice of mummified orange fruits of both ‘Tarocco Lempso’ and ‘Tarocco Tapi’. It is produced by several species of *Penicillium* [142]. Rubratoxin B is a mycotoxin exhibiting a range of acute and chronic toxic effects, including hepatotoxicity, nephrotoxicity, splenotoxicity, and genotoxicity. It is commonly found as contaminant in cereals, such as rice and wheat, and poses a potential health risk to humans and animals [126,143]. Lichexanthone is an additional secondary metabolite found in a considerable amount in the juice of mummified orange fruits. This is a well-known metabolite of lichens and was also found in many filamentous fungi, including endophytic *Penicillium* and *Trichoderma* species [144]. It has been reported to possess antimicrobial and antitumoral activity. Other secondary metabolites of *Penicillium*, recovered in orange fruits, included Alantrypinone, a metabolite with herbicidal activity [145]; Serantrypinone, a metabolite with insecticidal activity [146]; Atrovenetin, known for its antioxidant effects [114]; and Asteltoxin and Penochalasin K, both with proven antifungal activity [114,147]. Other *Penicillium* secondary metabolites with antimicrobial and antifungal activity such as Fungisporin, Palitantin, and Penipacid B were found in orange fruits. These metabolites were detected in both mummified and hail-injured fruits, and their presence seems correlated with the prevalence of *P. digitatum* over the other fungal species as inferred from the results of isolation. From an analogy with other plant pathogenic fungal species [148], it can be speculated that the ability of *P. digitatum* to produce several metabolites, which show allelopathic, antifungal, and antibacterial activity, might enhance the ecological fitness of this fungus. Moreover, this feature could explain the ability of *P. digitatum* to exclude or inhibit other less competitive species, such as *A. alternata* and *C. gloeosporioides*, from mummified fruits or to overgrow them during isolation of axenic cultures.

The PCA of the interactions among driving factors shaping the profile of secondary metabolites in orange fruits showed the profiles of mummified fruits were distinct from the profiles of either asymptomatic or hail-injured fruits. It is noteworthy the mycotoxin patulin was a characteristic component of the *Penicillium* metabolite profile of both the peel and juice of mummified fruits of both orange cultivars examined while the other *Penicillium*-related mycotoxin Rubratoxin B was detected with a high relative abundance only in the juice of this type of fruits. Moreover, the *Penicillium* metabolite profile from the peel of hail-injured fruits was distinct from the profile of asymptomatic fruits and was characterized by the presence of secondary metabolites, such as Alantrypinone, Anacine, Lichexanthone, Penipacid B, Serantrypinone, and Solistatine, whose toxic activity for humans and animals has been so far little investigated.

As for *C. gloeosporioides*, the metabolic profiles in both the peel and juice of hail-injured fruits of ‘Tarocco Lempso’ and ‘Tarocco Tapi’ grouped separately from the profiles of the corresponding asymptomatic fruits, suggesting an interaction between hail injuries and the biology of this fungus which is a common resident in the fruit peel but also a latent pathogen.

As for *A. alternata*, all the secondary metabolites identified in this study were recovered from mummified and hail-injured fruits. Only one of them, AME, has been reported as a mycotoxin of concern for human health produced by *Alternaria* species associated with citrus [24]. Little is known about the toxicological potential of the other *Alternaria*-related metabolites recovered from orange fruits in this study or the risk of fruit contamination. Most of the studies that were aimed to characterize the biological activity of these metabolites or the ability of fungal isolates to produce them were performed in vitro. In general, to evaluate the risk of food and feed contamination, it is noteworthy that overall 26 out of 47 metabolites identified in this study were recovered from both the fruit peel and the juice (6, 9, and 8 metabolites of *A. alternata*, *C. gloeosporioides*, and *P. digitatum*, respectively), 12 were recovered exclusively from the peel (4 per each of the three fungal species), and 12 were recovered exclusively from the juice (6 and 5 of *A. alternata* and *C. gloeosporioides*, respectively, 1 of *P. digitatum*). Consequently, the risk of contamination concerns both fresh fruits and processed products such as juices, marmalade, and citrus pulp, the last one used largely as a feed for cattle. 

## 4. Conclusions

Diverse secondary metabolites produced by three among the most common fungi responsible for pre- and post-harvest decays of citrus fruits were extracted directly from fruits sampled in commercial citrus orchards. They were identified using the modern analytical method of UHPLC–Q-TOF-MS. Many of these metabolites were isolated for the first time from citrus. The medium- and long-term objectives of this study were to evaluate the risk of contamination by mycotoxins and to provide basic information for setting maximum tolerance limits of mycotoxins in fresh citrus fruits and their derivatives. We are aware the latter is a more complex process and needs a multidisciplinary approach, as both the dietary habits of targeted populations and the additive effects of toxins from different food sources have to be taken into consideration. Regarding the risk of mycotoxin contamination of citrus fruits, an interesting finding is that with the analytical method we used a consistent level of secondary metabolites which were detected even in asymptomatic fruits. Moreover, the metabolic profile of the peel extract of asymptomatic fruits did not differ substantially from the profile of the peel extract of hail-damaged fruits. This is consistent with the results of isolations, which confirmed that *Alternaria*, *Colletotrichum*, and *Penicillium* reside in the peel of asymptomatic fruits as epiphytes, endophytes, or latent pathogens. UHPLC–Q-TOF-MS proved to be very effective as an analytical method for its rapidity, sensitivity, and robustness.

This study provided new insights into several theoretical and practical aspects concerning the biology and epidemiology of *A. alternata*, *C. gloeosporioides*, and *P. digitatum* and the identification of secondary metabolites that these fungi produce during the interaction with the host plant. In particular, this is the first study addressing the effects of hailstorms on fruit contamination by mycotoxins produced by fungi residing in the fruit peel as latent pathogens. In the light of the results of this study, the risk of contamination appears low for metabolites of *A. alternata* and *C. gloeosporioides*, even on fruits injured by the hail. Conversely, it seems real in fruits infected by *P. digitatum* as in both the peel and juice of decayed fruits we found considerable amounts of *Penicillium*-related toxins such as patulin and Rubratoxin B. Although mummified or severely blemished fruits usually do not enter the citrus fresh fruit supply chain as they are discarded before, this does not completely exclude the risk of contamination due to post-harvest rot. Moreover, mummified or severely blemished fruits are often present in bulk quantities destined for the juice industry. An accurate and severe selection of these fruits would prevent or reduce the risk of contamination by mycotoxins in the citrus juice industry.

## 5. Materials and Methods

### 5.1. Orange Fruit Sampling

On the 24 January 2022, mature fruits of two blood orange cultivars, ‘Tarocco Lempso’ and ‘Tarocco Tapi’, were collected in three 20-year-old commercial orchards cultivated with organic farming methods, in three diverse districts of the province of Catania (eastern Sicily). Around five months before the sampling, a hailstorm had affected the orchards, and since then trees had not been treated with fungicides. Three distinct types of fruits were selected per each cultivar: fruits injured by hail (fruits with necrotic lesions covering around 25% of the rind surface), mummified fruits still on the tree canopy, and asymptomatic fruits (fruits with no visible rind injury) (Figure 12).

Overall, in each orchard 60 fruits of each orange cultivar were collected. Five fruits of each type (hail-injured, mummified, and asymptomatic) were picked up randomly from four randomly selected trees. Overall, 60 fruits of the same type and cultivar were pooled together, and three replicates, each of 20 fruits, for each composite sample were processed and analyzed separately (Table 4). 

The sampling design is displayed in Table 4.

Fruit samples were kept into plastic bags and immediately shipped to the Laboratory of Food Toxicology at the Department of Preventive, Medicine, Nutrition and Food Science Area of the University of Valencia, Spain. Samples were stored at 4 °C until isolation of fungi from the rind. Subsequently, the juice was extracted, and secondary metabolites from the peel and juice were analyzed separately.

### 5.2. Isolation of Fungi

Isolations were conducted from the fruit peel in accordance with the method of Riolo et al. [17], with few modifications. Fruits were washed with tap water. Their surface was sterilized with a sodium hypochlorite solution (10%) for 1 min. Fruites were immersed in 70% ethanol for 30 s and rinsed in sterile distilled water (s.d.w.). After disinfection, 15 peel fragments (5 × 5 mm) per fruit were taken with a scalpel (Figure 13). In fruits damaged by hail, fragments were taken from necrotic lesions of the rind, while in mummified and asymptomatic fruits, fragments were taken randomly from the entire surface of the fruit. The fragments were blotted dry with sterile filter paper and placed in Petri dishes (three Petri dishes per fruit) on Potato Dextrose Agar (PDA; Oxoid Ltd., Basingstoke, UK) amended with 150 μg/mL streptomycin and incubated for 2 days at 25 °C, in the dark. Pure cultures of each fungal isolates were obtained by single-conidium subcultures.

### 5.3. Morphological Characterization of Isolates

The macroscopic characteristics (color, margin, diameter, and texture) of colonies were examined according to the method of Pryor and Michailides [149], whereas microscopic features (conidium and conidiophore branching morphology) were examined according to the method of Simmons [150] and Agrios [151] (Figure 13). Macro- and microscopic identification were used to separate the isolates by genus.

### 5.4. Molecular Characterization of Isolates

Genomic DNA was extracted from single-conidium isolates using the procedure described by Riolo et al. (2021) [17]. Overall, 290 selected fungal isolates of three distinct genera obtained from orange fruits were examined. Isolates were identified by the amplification and multilocal analysis of specific barcode regions (see Table 5). 

The PCR amplifications were performed on a GeneAmp PCR System 9700 (Applied Biosystems, Monza-Brianza, Italy). All PCRs were performed by using the Taq DNA polymerase recombinant (Invitrogen™, Carlsbad, 254 CA, USA) and carried out in a total volume of 25 µL containing the following: PCR Buffer (1X), dNTP mix (0.2 mM), MgCl2 (1.5 mM), forward and reverse primers (0.5 µM each), Taq DNA Polymerase (1 U), and 1 µL of genomic DNA (1 ng/µL) [34]. Amplified products were analyzed by electrophoresis, and single bands of the expected size were purified with the QIAquick PCR Purification Kit (Qiagen, Hilden, Germany) and sequenced with both forward and reverse primers by Colección Española de Cultivos tipo Parc Científic, Universitat de València (Valencia, España).

The identification was performed by comparing the sequences with other known sequences in the literature and present on the NCBI (https://blast.ncbi.nlm.nih.gov/Blast, accessed on 10 May 2023) and Mycobank (http://www.mycobank.org, accessed on 10 May 2023)/CBS-KNAW (Westerdijk Fungal Biodiversity Institute) (http://www.westerdijkinstitute.nl, accessed on 10 May 2023).

### 5.5. Preparation of Samples for the Analyses and Extraction of Orange Peel and Juice

Peel was manually separated from albedo using a knife, and the fresh orange juice was extracted from peeled and pressed oranges using a domestic squeezer. The peel and orange juice of each fruit were separated and individually transferred into conical polytetrafluoroethyl (PTFE) centrifuge tubes (15 mL). Before the extraction process, all the samples were stored in polyethylene tubes maintained at −80 °C.

### 5.6. Extraction of Fungal Secondary Metabolites from Orange Peel and Juice

The extraction of fungal secondary metabolites of orange peel was carried out in accordance with the method of Widiantini et al. [156], partially modified. At first, 5 mL of methanol per gram of orange peel was added into a conical polytetrafluoroethylene (PTFE) centrifuge tube (15 mL); then, it was shaken for 12 h at room temperature with an orbital laboratory shaker. The supernatant was collected and filtered through a 13-mm/0.22-μm nylon syringe filter (Membrane Solutions) in a 2 mL Amber Vial for chromatography analysis, and 20 μL of this solution was injected for UHPLC–Q-TOF-MS analysis.

The extraction of fungal secondary metabolites of orange juice was carried out according to the method of El Jai et al. [157], slightly modified. At first, 10 mL of orange juice and 20 mL of ethyl acetate (extraction solvent) were added in a conical polytetrafluoroethyl (PTFE) centrifuge tube (50 mL), and the suspension was mixed for 1 min; then, it was centrifuged for 10 min at 4000 rpm at 5 °C, using an Eppendorf centrifuge 5810R (Eppendorf, Hamburg, Germany). The organic phase at the top of the tube was recovered and placed in another PTFE centrifuge tube (15 mL) and saved. The total collected organic phase was evaporated to dryness under a nitrogen stream using a TurboVap LV evaporator (Zymark, Hopkinton, MA, USA). The dry extract was reconstituted with 1 mL of methanol and filtered through a 13-mm/0.22-μm nylon syringe filter (Membrane Solutions) in a 2 mL Amber Vial for chromatography analysis.

### 5.7. Analysis of Fungal Secondary Metabolites by UHPLC–Q-TOF-MS

Orange peel and orange juices extracts were injected for UHPLC–Q-TOF-MS analysis. All the analyses were performed in triplicate (*n* = 3). The samples were analyzed using an UHPLC device (1290 Infinity II LC, Agilent Technologies) composed of an automatic sampler, a binary pump, and a vacuum degasser, coupled with a quadrupole time of flight mass spectrometer (Agilent 6546 LC/Q-TOF), operating in positive ionization mode (Agilent Technologies, Santa Clara, CA, USA), which was used for chromatographic analysis. A total of 5 µL of sample was injected. Complete analysis was performed in 25 min. Chromatographic separation was performed with an Agilent Zorbax RRHD SB-C18, 2.1 mm × 50 mm, 1.8 µm column. Mobile phase A was composed of Milli-Q water, and acetonitrile was used for mobile phase B (both phases were acidified with 0.1% formic acid), with gradient elution, as follows: 0 min, 2% B; 22 min 95% B; and 25 min, 5% B. The column was equilibrated for 3 min before every analysis. The flow rate was 0.4 mL/min. Dual AJS ESI source conditions were as follows: gas temperature: 325 C; gas flow: 10 L/min; nebulizer pressure: 40 psig; sheath gas temperature: 295 C; sheath gas flow: 12 L/min; capillary voltage: 4000 V; nozzle voltage: 500 V; Fragmentor: 120 V; skimmer: 70 V; product ion scan range: 100–1500 Da; MS scan rate: 5 spectra/s; MS/MS scan rate: 3 spectra/s; maximum precursors per cycle: 2; and collision energy: 10, 20, and 40 eV. An untargeted LC/Q-TOF-based metabolomics approach was used to identify the different secondary metabolites of fungi growing on orange peel and orange juice. Integration, data elaboration, and identification of metabolites were managed using MassHunter Qualitative Analysis Software B.08.00 and library PCDL Manager B.08.00 [85].

### 5.8. Statistical Analysis

The analytical data obtained by UHPLC–Q-TOF-MS were log10 transformed before statistical analysis. Relationships among two different cultivars (‘Tarocco Lempso’ and ‘Tarocco Tapi’) × *Alternaria*, *Colletotrichum*, and *Penicillium* species combinations were analyzed using Pearson’s correlation coefficient analysis. All the above statistical analyses were performed using RStudio v.1.2.5 (R). MetaboAnalyst 5.0 software [158] was used for principal component analysis (PCA) using log10 transformed data. The features included were log-transformed and mean-centered.

## Figures and Tables

**Figure 1 toxins-15-00407-f001:**
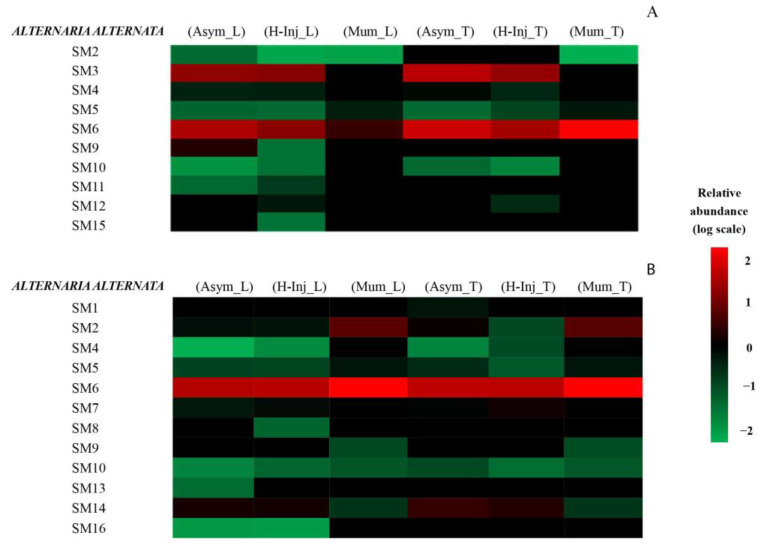
Heat map representing the relative abundance of *Alternaria alternata* secondary metabolites detected in orange peel (**A**) and orange juices (**B**). Colors are based on the relative abundance (logarithmic scale) of the secondary metabolites detected, where red represents high abundance and green represents low abundance.

**Figure 2 toxins-15-00407-f002:**
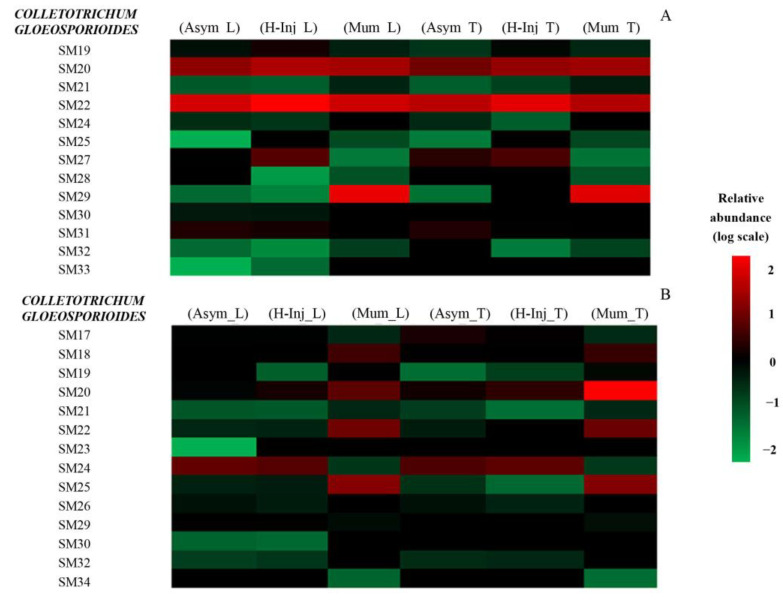
Heat map representing the relative abundance of *Colletotrichum gloeosporioides* secondary metabolites detected in orange peel (**A**) and orange juices (**B**). Colors represent the relative abundance (logarithmic scale) of the secondary metabolites: red represents high abundance, and green represents low abundance.

**Figure 3 toxins-15-00407-f003:**
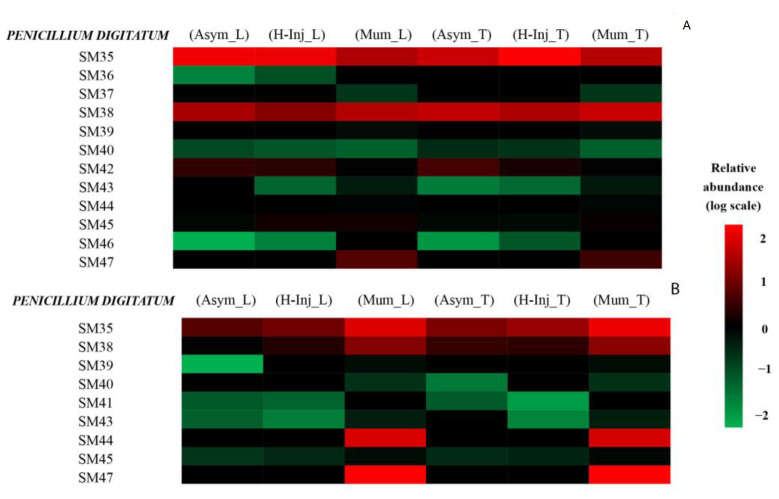
Heat map representing the relative abundance of *Penicillium digitatum* secondary metabolites detected in orange peel (**A**) and orange juices (**B**). Colors represent the relative abundance (logarithmic scale) of the secondary metabolites detected: red represents high abundance, and green represents low abundance.

**Figure 4 toxins-15-00407-f004:**
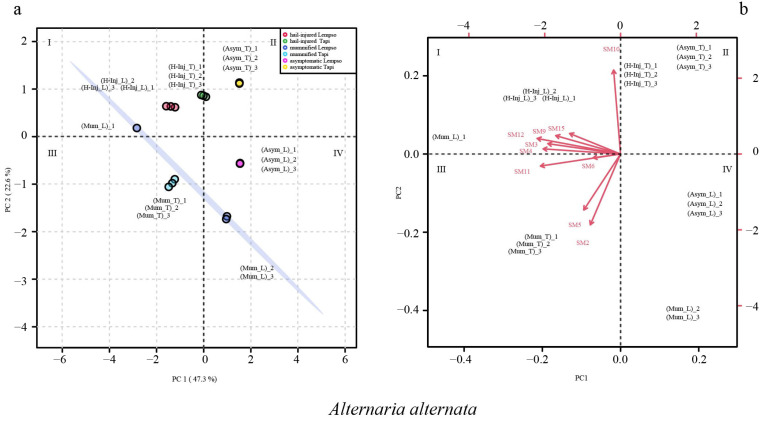
Principal Component Analysis (PCA), scores plot (**a**) and biplot (**b**), based on secondary metabolites of *Alternaria alternata* detected in the peel of ‘Tarocco Lempso’ (L) and ‘Tarocco Tapi’ (T) fruits, with three distinct replicates per each type of fruit, asymptomatic (Asym), hail-injured (H-Inj), and mummified (Mum).

**Figure 5 toxins-15-00407-f005:**
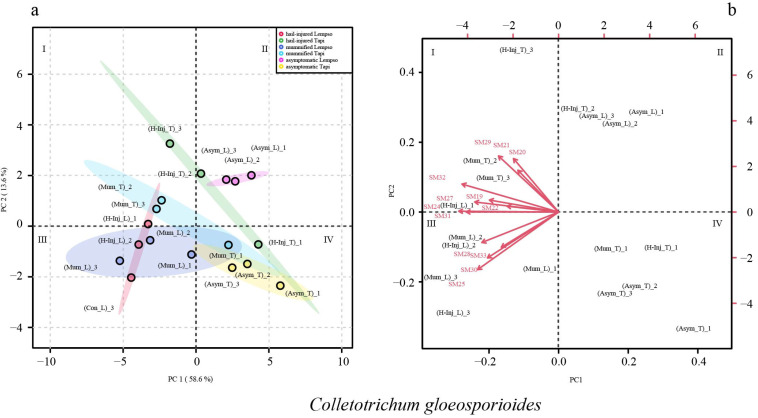
Principal Component Analysis (PCA), scores plot (**a**) and biplot (**b**), based on secondary metabolites of *Colletotrichum gloeosporioides* detected in the peel of ‘Tarocco Lempso’ (L) and ‘Tarocco Tapi’ (T) fruits, with three distinct replicates per each type of fruit, asymptomatic (Asym), hail-injured (H-Inj), and mummified (Mum).

**Figure 6 toxins-15-00407-f006:**
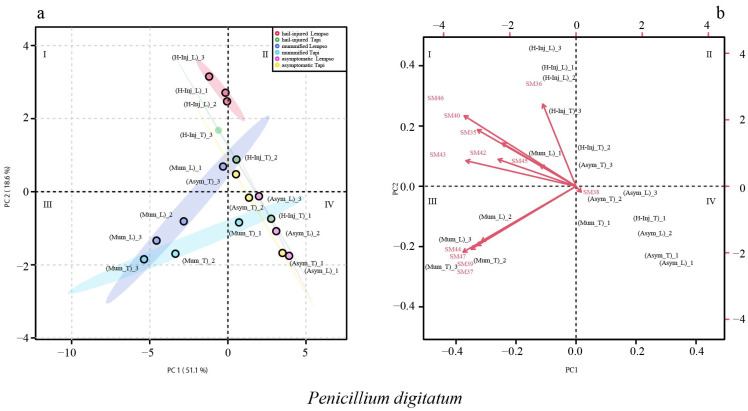
Principal Component Analysis (PCA), scores plot (**a**) and biplot (**b**), based on secondary metabolites of *Penicillium digitatum* detected in the peel of ‘Tarocco Lempso’ (L) and ‘Tarocco Tapi’ (T) fruits, with three distinct replicates per each type of fruit, asymptomatic (Asym), hail-injured (H-Inj), and mummified (Mum).

**Figure 7 toxins-15-00407-f007:**
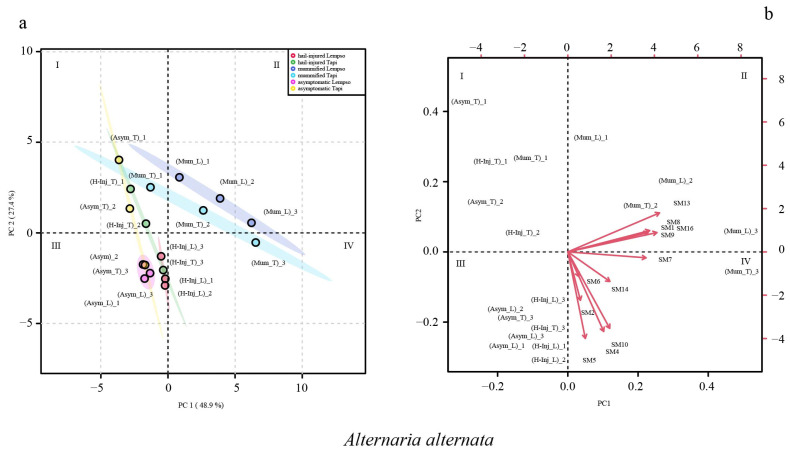
Principal Component Analysis (PCA), scores plot (**a**) and biplot (**b**), based on secondary metabolites of *Alternaria alternata* detected in the juice of ‘Tarocco Lempso’ (L) and ‘Tarocco Tapi’ (T) fruits, with three distinct replicates per each type of fruit, asymptomatic (Asym), hail-injured (H-Inj), and mummified (Mum).

**Figure 8 toxins-15-00407-f008:**
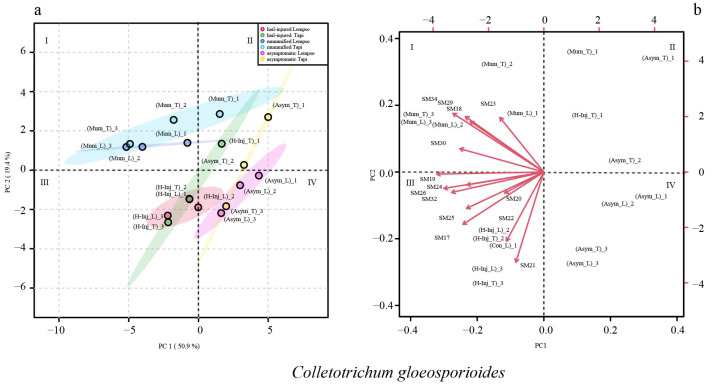
Principal Component Analysis (PCA), scores plot (**a**) and biplot (**b**), based on secondary metabolites of *Colletotrichum gloeosporioides* detected in the juice of ‘Tarocco Lempso’ (L) and ‘Tarocco Tapi’ (T) fruits, with three distinct replicates per each type of fruit, asymptomatic (Asym), hail-injured (H-Inj), and mummified (Mum).

**Figure 9 toxins-15-00407-f009:**
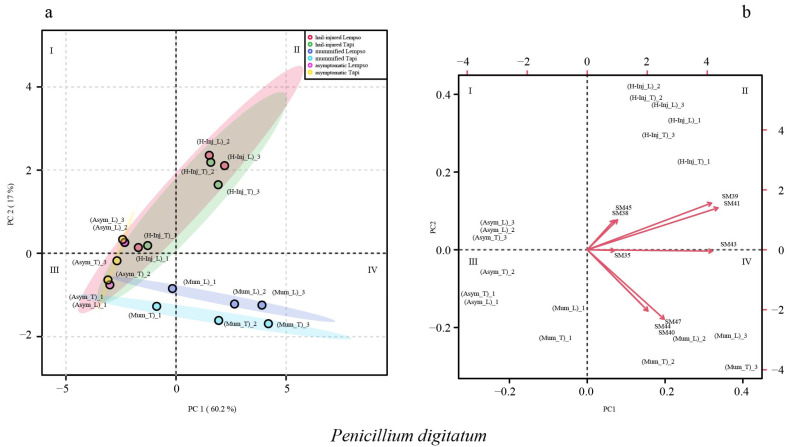
Principal Component Analysis (PCA), scores plot (**a**) and biplot (**b**), based on secondary metabolites of *Penicillium digitatum* detected in the juice of ‘Tarocco Lempso’ (L) and ‘Tarocco Tapi’ (T) fruits, with three distinct replicates per each type of fruit, asymptomatic (Asym), hail-injured (H-Inj), and mummified (Mum).

**Figure 10 toxins-15-00407-f010:**
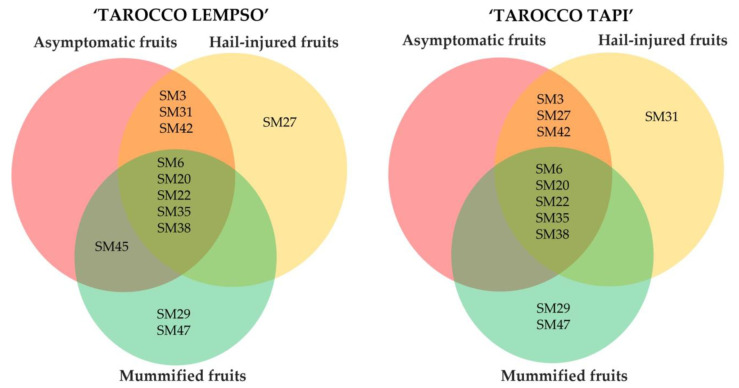
Venn diagrams showing the distribution of secondary metabolites in peel extracts of hail-injured, mummified, and asymptomatic fruits of ‘Tarocco Lempso’ and ‘Tarocco Tapi’ cultivars.

**Figure 11 toxins-15-00407-f011:**
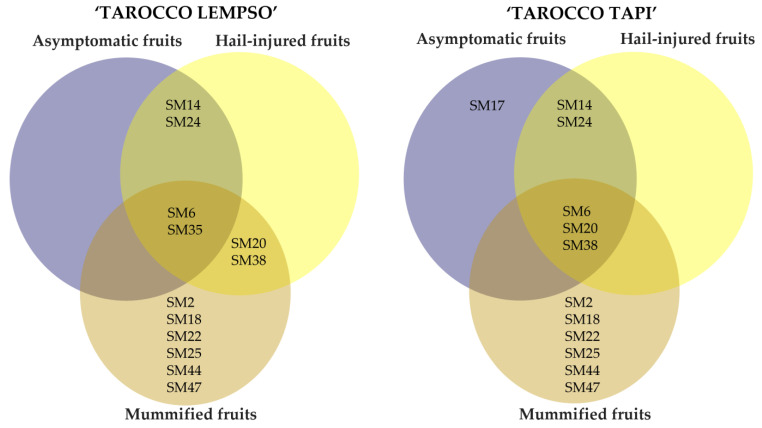
Venn diagrams showing the distribution of secondary metabolites in juice of hail-injured, mummified, and asymptomatic fruits of ‘Tarocco Lempso’ and ‘Tarocco Tapi’ cultivars.

**Figure 12 toxins-15-00407-f012:**
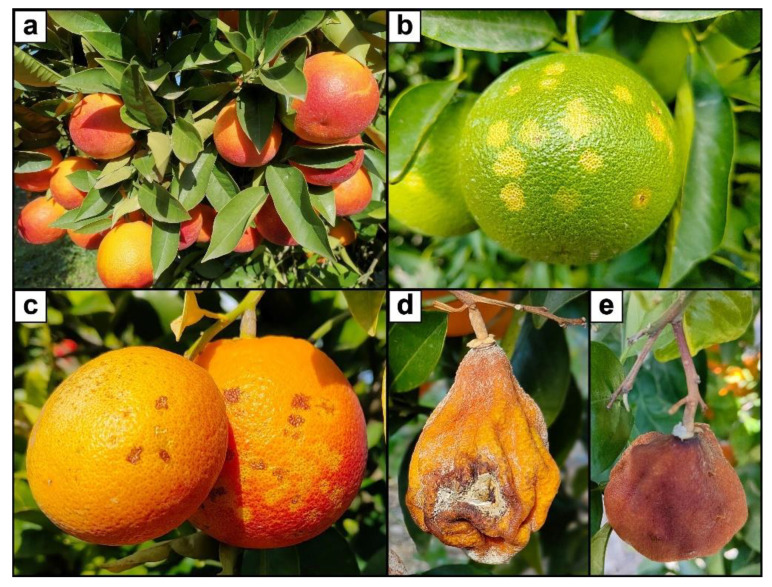
(**a**) Asymptomatic mature fruits of ‘Tarocco Lempso’; (**b**) Rind blemishes (oleocellosis) on a green fruit in autumn, few days after the hail-storm; (**c**) Blemished (hail-injured) mature fruits of ‘Tarocco Lempso’; (**d**,**e**) Mummified fruit of ‘Tarocco Lempso’.

**Figure 13 toxins-15-00407-f013:**
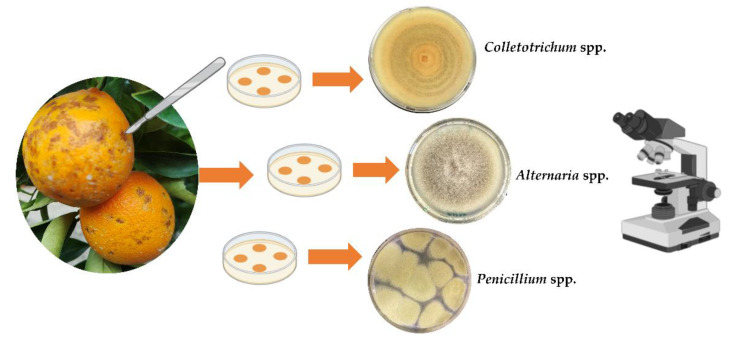
Diagrammatic representation of isolation and morphological identification at genus level of isolates obtained from orange fruits.

**Table 1 toxins-15-00407-t001:** Proportion of isolates of three fungal species from three different types of orange fruits subdivided into proportion of isolates per each cluster.

Pathogen	Types of Fruit	Cluster	Rate of Pathogens Isolated (%)	Total Rate (%)
** *Alternaria alternata* **	hail-injured	hail-injured ‘Lempso’	5%	12%
		hail-injured ‘Tapi’	4%	
	mummified	mummified ‘Lempso’	0%	
		mummified ‘Tapi’	0%	
	asymptomatic	asymptomatic ‘Lempso’	1%	
		asymptomatic ‘Tapi’	2%	
** *Colletotrichum* ** ** *gloeosporioides* **	hail-injured	hail-injured ‘Lempso’	5%	28%
		hail-injured ‘Tapi’	7%	
	mummified	mummified ‘Lempso’	2%	
		mummified ‘Tapi’	2%	
	asymptomatic	asymptomatic ‘Lempso’	7%	
		asymptomatic ‘Tapi’	5%	
** *Penicillium digitatum* **	hail-injured	hail-injured ‘Lempso’	10.5%	60%
		hail-injured ‘Tapi’	10%	
	mummified	mummified ‘Lempso’	15.5%	
		mummified ‘Tapi’	15%	
	asymptomatic	asymptomatic ‘Lempso’	4%	
		asymptomatic ‘Tapi’	5%	

**Table 2 toxins-15-00407-t002:** Isolates of *Alternaria*, *Colletotrichum*, and *Penicillium* characterized in this study at species level.

Isolate Code	Species	Accession Number			
		ITS-rDNA	ß-tubulin 2	TEF-1a	Alt-1a
ID8O4	*Alternaria alternata*	OQ929058	-	OQ939919	OQ938706
ID8O5	*Alternaria alternata*	OQ929059	-	OQ939920	OQ938707
ID8O6	*Alternaria alternata*	OQ929060	-	OQ939921	OQ938708
ID8O7	*Alternaria alternata*	OQ929061	-	OQ939922	OQ938709
ID8O8	*Alternaria alternata*	OQ929062	-	OQ939923	OQ938710
ID8O9	*Alternaria alternata*	OQ929063	-	OQ939924	OQ938711
ID8O10	*Alternaria alternata*	OQ929064	-	OQ939925	OQ938712
ID8O11	*Alternaria alternata*	OQ929065	-	OQ939926	OQ938713
ID8O12	*Alternaria alternata*	OQ929066	-	OQ939927	OQ938714
ID3O13	*Alternaria alternata*	OQ929067	-	OQ939928	OQ938715
ID3O14	*Alternaria alternata*	OQ929068	-	OQ939929	OQ938716
ID3O15	*Alternaria alternata*	OQ929069	-	OQ939930	OQ938717
ID3O16	*Alternaria alternata*	OQ929070	-	OQ939931	OQ938718
ID3O17	*Alternaria alternata*	OQ929071	-	OQ939932	OQ938719
ID3O18	*Alternaria alternata*	OQ929072	-	OQ939933	OQ938720
ID3O19	*Alternaria alternata*	OQ929073	-	OQ939934	OQ938721
ID3O20	*Alternaria alternata*	OQ929074	-	OQ939935	OQ938722
ID3O21	*Alternaria alternata*	OQ929075	-	OQ939936	OQ938723
ID3O22	*Alternaria alternata*	OQ929076	-	OQ939937	OQ938724
ID3O23	*Alternaria alternata*	OQ929077	-	OQ939938	OQ938725
ID5O12	*Colletotrichum gloeosporioides*	OQ929078	OQ938686	-	-
ID5O13	*Colletotrichum gloeosporioides*	OQ929079	OQ938687	-	-
ID5O14	*Colletotrichum gloeosporioides*	OQ929080	OQ938688	-	-
ID5O15	*Colletotrichum gloeosporioides*	OQ929081	OQ938689	-	-
ID5O16	*Colletotrichum gloeosporioides*	OQ929082	OQ938690	-	-
ID5O17	*Colletotrichum gloeosporioides*	OQ929083	OQ938691	-	-
ID5O18	*Colletotrichum gloeosporioides*	OQ929084	OQ938692	-	-
ID5O19	*Colletotrichum gloeosporioides*	OQ929085	OQ938693	-	-
ID1O20	*Colletotrichum gloeosporioides*	OQ929086	OQ938694	-	-
ID1O21	*Colletotrichum gloeosporioides*	OQ929087	OQ938695	-	-
ID1O22	*Colletotrichum gloeosporioides*	OQ929088	OQ938696	-	-
ID1O23	*Colletotrichum gloeosporioides*	OQ929089	OQ938697	-	-
ID1O24	*Colletotrichum gloeosporioides*	OQ929090	OQ938698	-	-
ID1O25	*Colletotrichum gloeosporioides*	OQ929091	OQ938699	-	-
ID1O26	*Colletotrichum gloeosporioides*	OQ929092	OQ938700	-	-
ID1O27	*Colletotrichum gloeosporioides*	OQ929093	OQ938701	-	-
ID1O28	*Colletotrichum gloeosporioides*	OQ929094	OQ938702	-	-
ID1O29	*Colletotrichum gloeosporioides*	OQ929095	OQ938703	-	-
ID1O30	*Colletotrichum gloeosporioides*	OQ929096	OQ938704	-	-
ID1O31	*Colletotrichum gloeosporioides*	OQ929097	OQ938705	-	-
ID4O5	*Penicillium digitatum*	OQ929098	OQ938666	-	-
ID4O6	*Penicillium digitatum*	OQ929099	OQ938667	-	-
ID4O7	*Penicillium digitatum*	OQ929100	OQ938668	-	-
ID4O8	*Penicillium digitatum*	OQ929101	OQ938669	-	-
ID4O9	*Penicillium digitatum*	OQ929102	OQ938670	-	-
ID4O10	*Penicillium digitatum*	OQ929103	OQ938671	-	-
ID4O11	*Penicillium digitatum*	OQ929104	OQ938672	-	-
ID4O12	*Penicillium digitatum*	OQ929105	OQ938673	-	-
ID4O13	*Penicillium digitatum*	OQ929106	OQ938674	-	-
ID4O14	*Penicillium digitatum*	OQ929107	OQ938675	-	-
ID4O15	*Penicillium digitatum*	OQ929108	OQ938676	-	-
ID4O16	*Penicillium digitatum*	OQ929109	OQ938677	-	-
ID4O17	*Penicillium digitatum*	OQ929110	OQ938678	-	-
ID4O18	*Penicillium digitatum*	OQ929111	OQ938679	-	-
ID10O4	*Penicillium digitatum*	OQ929112	OQ938680	-	-
ID10O5	*Penicillium digitatum*	OQ929113	OQ938681	-	-
ID10O6	*Penicillium digitatum*	OQ929114	OQ938682	-	-
ID10O7	*Penicillium digitatum*	OQ929115	OQ938683	-	-
ID10O8	*Penicillium digitatum*	OQ929116	OQ938684	-	-
ID10O9	*Penicillium digitatum*	OQ929117	OQ938685	-	-

**Table 3 toxins-15-00407-t003:** Secondary metabolites of *Alternaria alternata*, *Colletotrichum gloeosporioides*, and *Penicillium digitatum* identified in this study in the peel and juice of blood orange fruits.

Secondary Metabolites	ID Code ^a^	Matrices ^b^	Chemical Class	Reference ^c^
AAL-toxin TB2	SM1	juice	Mycotoxins (Cyclic peptides)	[88]
AAL-toxin TE2	SM2	peel and juice	Mycotoxins	[88]
AF-toxin II	SM3	peel	Mycotoxins (Fumonisins)	[89]
Altenusin (ALN)	SM4	peel and juice	Benzoxazole	[58]
Alternariol monomethyl ether (AME)	SM5	peel and juice	Mycotoxins	[90]
Alternethanoxin A	SM6	peel and juice	Ketone	[91]
Altersolanol L	SM7	juice	Derivatives of anthraquinone	[92]
Aurasperone C	SM8	juice	Prenylated benzophenones	[93]
Curvularin	SM9	peel and juice	Macrocyclic lactones	[94]
Dihydroaltersolanol	SM10	peel and juice	Cyclohexenones	[95]
Erythroglaucin	SM11	peel	Anthraquinones	[88]
Macrosporin A	SM12	peel	Macrolides	[58]
Maculosin	SM13	juice	Tetramic acids	[96]
Porriolide	SM14	juice	Furanones	[97]
Porritoxinol	SM15	peel	Phthalides	[98]
Tentoxin (TEN)	SM16	juice	Cyclic tetrapeptides	[58]
(+)-(3R,4S)-cis-4-hydroxy-6-deoxyscytalone	SM17	juice	Tetramic acids	[99]
Ergosterol peroxide	SM18	juice	Steroids	[100]
5,4-dihydroxy-3,6,7-trimethoxy-8C-methylflavone	SM19	peel and juice	Flavones	[101]
5,4-dihydroxy-3,7,8-trimethoxy-6C-methylflavone	SM20	peel and juice	Flavones	[101]
Alternariol-5-Me ether	SM21	peel and juice	Coumarins	[102]
Apigenin-8-C-β-D-glucopyranoside	SM22	peel and juice	Flavonoid glycosides	[103]
Collectotrichin A	SM23	juice	Cyclopeptides	[104]
Colletofragarone A1	SM24	peel and juice	Sesquiterpenoids	[105]
Colletoic acid	SM25	peel and juice	Diketopiperazines	[106]
Colletoketol	SM26	juice	Macrolides	[101]
Colletomelleins B	SM27	peel	Quinones	[107]
Colletonoic acid	SM28	peel	Carboxylic acids	[108]
Colletotrilactam A	SM29	peel and juice	Lactams	[109]
Colletotrilactam D	SM30	peel and juice	Lactams	[109]
Fusarentin 6,7-dimethyl ether	SM31	peel	Coumarins	[110]
Novae-zelandin A	SM32	peel and juice	Phenazines	[111]
Phthalide	SM33	peel	Cyclic anhydrides	[110]
Pyrenocine A	SM34	juice	Pyrenocines	[112]
Alantrypinone	SM35	peel and juice	Quinazolinones	[88]
Anacine	SM36	peel	Quinazolinones	[113]
Asteltoxin	SM37	peel	Anthraquinones	[114]
Atrovenetins	SM38	peel and juice	Xanthones	[114]
Fungisporin	SM39	peel and juice	Cyclohexenones	[115]
Lichexanthone	SM40	peel and juice	Xanthones	[116]
Palitantin	SM41	juice	Flavones	[117]
Penipacid B	SM42	peel	Diketopiperazines	[118]
Penochalasin K	SM43	peel and juice	Macrolides	[119]
Rubratoxin B	SM44	juice	Mycotoxins	[120]
Serantrypinone	SM45	peel and juice	Quinones	[121]
Solistatin	SM46	peel	Lactones	[122]
Patulin	SM47	peel and juice	Mycotoxins (Furanocoumarins)	[46]

^a^ SM1-SM16: *Alternaria alternata*; SM17-SM34: *Colletotrichum gloeosporioides*; SM35-SM47 *Penicillium digitatum*. ^b^ Matrices on which the secondary metabolites were identified in this study. ^c^ Original reports of the secondary metabolites produced by *Alternaria*, *Colletotrichum*, and *Penicillium* spp.

**Table 4 toxins-15-00407-t004:** Sampling and processing design for the determination of metabolite profile of fruits of two blood orange cultivars.

Cluster	Number of Fruits	Cultivar
Asymptomatic fruits (Asym_L)	60 ^a^	Tarocco Lempso
Hail-injured fruits (H-Inj_L)	60	
Mummified fruits (Mum_L)	60	
Asymptomatic fruits (Asym_T)	60	Tarocco Tapi
Hail-injured fruits (H-Inj_T)	60	
Mummified fruits (Mum_T)	60	

^a^ Three replicates, each of 20 fruits.

**Table 5 toxins-15-00407-t005:** Barcode region/gene, primers, expected amplicon size (bp), and related references of the fungal isolates identified in this study.

Barcode Region/Gene	Target Organism/s	Primers	Expected Amplicon Size (bp)	References
Internal transcribed spacer (ITS) region of the ribosomal DNA	*Alternaria*, *Colletotrichum*, and *Penicillium*	ITS-1 5′-TCCGTAGGTGAACCTGCGG-3′	~700	[152]
		ITS-4 5′-TCCTCCGCTTA TTGATATGC-3′		
Intron of translation elongation factor 1 alpha gene (*tef1*)	*Alternaria*	EF1-728F 5′-CAT CGA GAA GTT CGA GAA GG-3′	~350	[153]
		EF1-986R 5′-TAC TTG AAG GAA CCC TTA CC-3′		
β-tubulin gene (*tub2*)	*Colletotrichum* and *Penicillium*	Bt2a 5′-GGTAACCAAATCGGTGCTGCTTTC-3′	~400 (*Penicillium*) ~773 (*Colletotrichum*)	[154]
		Bt2b 5′-ACCCTCAGTGTAGTGACCCTTGGC-3′		
Alt-1a allergen protein	*Alternaria*	Alt-for 5′-TATGCAGTTCACCACCATCGC-3′	~400	[155]
		Alt-rev 5′-ACGAGGGTGAYGTAGGCGTC-3′		

## Data Availability

Data will be available upon specific request to the authors.

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
