# Peer review of "Diversity of Mycotoxins and Other Secondary Metabolites Recovered from Blood Oranges Infected by Colletotrichum, Alternaria, and Penicillium Species"

_toxins, 2023, doi:10.3390/toxins15070407_

Round 1

Reviewer 1 Report

The manuscript “Diversity of mycotoxins and other secondary metabolites recovered from blood oranges infected by Colletotrichum, Alternaria, and Penicillium species” aims to identify mycotoxins produces by A. alternata, C. gloeosporioides and P. digitatum in three types of fruits (injured by hail, mummified and asymptomatic) of two blood orange cultivars “Tarocco Lembo” and “Tarocco Tapi”. To identify mycotoxins the analytical methods of UHPLC-Q-TOF-MS was used.

The authors state that this is the first study to deal with effects of hailstorms on fruit contamination by mycotoxins produced by fungi residing in the fruit peel as latent pathogens. They concluded the risk of contamination appears low for metabolites of A. alternata and C. gloeosporioides, while it seems real in fruits infected by P. digitatum, in fact they found considerable amounts of Penicillium related toxins such as Patulin and Rubratoxin B.

The study's context is thoroughly explained in the introduction section. There is enough supporting literature provided to provide the reader with all the knowledge they need to understand the study's topic.

The study's conclusions are largely supported by the data and information provided by the authors, and it was well-organized, produced new data. 

Here are the points I think should be corrected:

Lines 410-411: why asymptomatic is identify with “NoCon” and hail injured with “Con”? These abbreviations are not present in the figures. Please, correct.

Figure 8: The abbreviations “NoCon” and “Con” are used in figures a and b. It is correct?

Line 458: Please write “secondary” not in italics.

Line 699: Tarocco “Lempo”? Please, correct.

Line 730: Please, add point after Figure 13 and eliminate “–“

Line 735: Please add space between “[152]” and  “and”

Line 741: Please, add the number of reference “Riolo et al., 2021”

Line 739: In Section 5.4, I suggest adding a table containing the primers information (name, sequences, target, amplicon size and references) used for isolates characterization  

Lines 753-754: 1 µL of what? Please, specify the concentration in ng of DNA used in PCR.

Line 757: Please change “(“ before “Valencia” with comma

Line 773: Please, correct the reference number “[1589]”

Line 789: Please add space between “2“ and “mL”

In References:

Reference 38: Please, delete “(2010)” after authors list

Reference 43: Please, delete “(2009)” after authors list

Reference 45: Please, delete “(2017)” after authors list and write Journal name in abbreviated form

Reference 64: Please, delete “(2017)” after authors list

Reference 93: Please, delete “()” after authors list

Reference 98: Please, delete “()” after authors list

Reference 108: Please, complete the first author name

Reference 119: Please, delete “(2021)” after authors list

Reference 139: Please, delete “&” before the last author

Reference 150: Please, write Journal name in italics

Reviewer 2 Report

The paper is well written, representing a big base of important databases. Please read the following several comments:

The aim of this study was to show the diversity of mycotoxins and other secondary in blood oranges. Therefore, lines 134-145 are not necessary. These sentences are well-known facts.

Lines 773: 158 or 159 references?

Line 772: The extraction of fungal secondary metabolites; Line 780: The extraction of fungal metabolites, please use the same terminology.

Extraction for fungal secondary metabolites from orange peel included shaking for 12 hours, but extraction for fungal secondary metabolites from orange juice included mixing of 1 min? Please explain these significant differences.

The authors wrote “Preparation of samples for the analyses and extraction of orange juice”, however “Preparation of samples for the analyses and extraction of orange peel”, is missing.

There is no information about the validation of applied UHPLC– Q-TOF-MS. For a high-quality scientific paper, this information is necessary, especially due to the complexity of matrixes such as orange juice and orange peel.

Results are very huge, presented on even 16 pages. However, results were presented by different statistical analyses, and in that way, there are not very readable. It is a little bit difficult to find the most important results and main finding of this paper on 16 pages. It should be written in a more readable manner.

Minor editing of English language required
